# Development of an orally-administrable tumor vasculature-targeting therapeutic using annexin A1-binding D-peptides

**Motohiro Nonaka[1,2], Hideaki Mabashi-Asazuma[3], Donald L. Jarvis[3], Kazuhiko Yamasaki[4], Tomoya O. Akama[5], Masato Nagaoka[6], Toshio Sasai[6], Itsuko Kimura-Takagi[6], Yoichi Suwa[6], Takashi Yaegashi[6], Chun-Teng Huang[7], Chizuko Nishizawa-Harada[1], Michiko N. Fukuda[1,7] ***

1 Laboratory for Drug Discovery, National Institute of Advanced Industrial Science and Technology, Tsukuba, Ibaraki, Japan, 2 Department of Biological Chemistry, Human Health Sciences, Graduate School of Medicine, Kyoto University, Kyoto, Japan, 3 Department of Molecular Biology, University of Wyoming, Laramie, WY, United States of America, 4 Biomedical Research Institute, National Institute of Advanced Industrial Science and Technology, Tsukuba, Ibaraki, Japan, 5 Department of Pharmacology, Kansai Medical University, Hirakata, Osaka, Japan, 6 Yakult Central Institute, Kunitachi, Tokyo, Japan, 7 Cancer Center, Sanford-Burnham-Prebys Medical Discovery Institute, La Jolla, CA, United States of America

* michiko@sbpdiscovery.org

**Data Availability Statement:** All relevant data are within the manuscript and its Supporting Information files.

## Abstract

We previously reported that IF7 peptide, which binds to the annexin A1 (ANXA1) N-terminus, functions as a tumor vasculature-targeted drug delivery vehicle after intravenous injection. To enhance IF7 stability *in vivo*, we undertook mirror-image peptide phage display using a synthetic D-peptide representing the ANXA1 N-terminus as target. We then identified peptide sequences, synthesized them as D-amino acids, and designated the resulting peptide dTIT7, which we showed bound to the ANXA1 N-terminus. Whole body imaging of mouse brain tumor models injected with near infrared fluorescent IRDye-conjugated dTIT7 showed fluorescent signals in brain and kidney. Furthermore, orally-administered dTIT7/geldanamycin (GA) conjugates suppressed brain tumor growth. Ours is a proof-of-concept experiment showing that ANXA1-binding D-peptide can be developed as an orally-administrable tumor vasculature-targeted therapeutic.

## Introduction

It is widely accepted that surfaces of the vasculature are heterogenous and express varying tissue-specific receptors under different pathological conditions [1]. Oh *et al.* used subtractive proteomics analysis of malignant vs. normal vasculature to identify Annexin A1 (ANXA1) as highly specific surface marker of malignant tumor vasculature [2, 3]. Independent of these studies, we looked for carbohydrate mimetic peptides to inhibit cell surface carbohydrate-dependent hematogenous cancer colonization of the lung [4–6]. Since carbohydrate antigen specificity is determined by 3–4 carbohydrate residues of 600–800 Da, we assumed that a 7-mer peptide of 770 Da would mimic a carbohydrate antigen. We screened a phage library of linear 7-mer peptides using monoclonal anti-LewisA antibody and obtained a series of

**Funding:** This study was supported by an institutional grant LEAD at National Institute of Advanced Industrial Science and Technology, by the Project for Cancer Research and Therapeutic Evolution (PCREATE) from the Japan Agency for Medical Research and Development (AMED) to MNF, and by P41 GM103390 and P41 RR005351. NM is a recipient of a Research Grant for Young Japanese Scientists from The Nakajima Foundation.

**Competing interests:** The authors have declared that no competing interests exist.

peptides with the consensus sequence IXLLXXR [5]. Subsequent studies showed that these peptides bound to endothelial surface receptors, including ANXA1 [7, 8]. Finally, we found that a linear 7-mer peptide IFLLWQR (IF7) binds to the ANXA1 N-terminus [7–9]. Upon intravenous injection into tumor-bearing mice, a conjugate of IF7 with the anti-cancer drug geldanamycin (GA) suppressed growth of prostate, breast, melanoma and lung tumors, and IF7-conjugated SN-38 suppressed colon cancer growth in mice at low dose without side effects [8]. Equally significantly, we found intravenously-injected IF7 accumulated on the tumor endothelial cell surface, was endocytosed into vesicles, and crossed tumor endothelial cells by transcytosis [8]. Based on these observations we hypothesized that an IF7-conjugated anti-cancer drug might be able to overcome the blood-brain barrier (BBB) and eradicate brain tumors. Indeed, intravenous injection of the IF7 conjugated to the anti-tumor agent SN-38 into model mice harboring brain tumors efficiently reduced the size of those tumors, even at low dosage, and apparently invoked a host immune reaction against brain tumors, leading to complete remission [9].

IF7 is conjugated to SN-38 through an esterase-cleavable linker, allowing release of SN-38 from the peptide once it reaches the tumor vasculature. IF7 peptide itself is also susceptible to proteases, compromising IF7-SN38 stability *in vivo* [8]. To circumvent stability issues, many investigators cyclize L-peptides to enhance protease-resistance [10]. Others address stability issues by designing D-peptides or retro-inverso forms peptides in which L- amino acids are replaced with D-amino acids and the primary sequence of the peptide reversed [11]. However, a retro-inverso form of IF7 (designated RIF7) appears to have targeted normal organs nonspecifically [12]. We were also concerned about IF7 or RIF7 hydrophobicity. We therefore sought alternate ways to create a more stable and less hydrophobic form of IF7.

Synthetic peptides from the ANXA1 N-terminal domain exhibit biological activity in several models of experimental inflammation [13, 14]. We tested a series of synthetic ANXA1 N-terminal peptides and found that the N-terminal 1–15 residues are sufficient for IF7 binding [9]. Here, to construct a protease-resistant form of IF7 that retains ANXA1-binding activity, we undertook mirror-image phage library screening [15–17], taking advantage of the fact that IF7 binds to the chemically synthesized ANXA1 N-terminus (1–15 residues plus a cysteine at 16), a peptide that we previously designated MC16 [9]. This screen identified the peptide dTIT7 as a D-type peptide that binds the ANXA N-terminus. We then conjugated dTIT7 to geldanamycin (GA) through an uncleavable linker to generate GA-dTIT7.

Orally-administrable therapeutics decrease pain for patients and provide their caregivers a simpler method to dispense treatment. As a technical challenge of peptide therapeutics is to remain stable *in vivo*, and GA-dTIT7 is anticipated to satisfy this condition, we tested bioactivity of orally-administered GA-dTIT7 on mouse brain tumor models. We present proof-of-concept data showing that orally-administered GA-dTIT7 suppresses brain tumor growth in mice.

## Materials and methods

### Materials

Unless noted, peptides used here, including the D-type peptides dTIT7 (TITWPTM), dLRF7 (LRFPTVL), dSPT7 (SPTSLLF), dLKG7 (LKGMLRI) and dLLS7 (LLSWPSA), were synthesized by GenScript (Piscataway, NJ). D-MC16 and L-MC16 peptides, which consist of 15 N-terminal residues of human ANXA1 plus a cysteine residue at 16 position (MAMVSEFLK-QAWFIEC), and L-MC16 mutants were synthesized by Bio-Synthesis (Lewisville, TX). *Iso*-dTIT7, in which prolines contain $^{13}C$ and $^{15}N$, were synthesized by Peptide Institute, Osaka, Japan. Luciferin sodium salt (VivoGlo) was from Promega.

## Mirror-image phage library screening

Library screening strategies [15, 16] were adapted to identify a peptide sequence binding the ANXA1 N-terminal domain. D-MC16 peptide (described above) was dissolved in DMSO and used to coat maleimide-activated plates (Corning) at 10 nmol/well at 4°C for 20 hours. After blocking with SuperBlock solution (Thermo), screening was performed using a T7 phage library comprised of fully random 7-mer peptides and of $20^7 = 1.28 \times 10^9$ diversity, provided by Dr. E. Ruoslahti, Sanford-Burnham-Prebys Medical Discovery Institute (SBP). The phage peptide sequence was determined using an Ion Torrent Next Generation sequencer (Thermo). Top-ranked sequences TITWPTM (dTIT7) and the next four high-ranking peptides were then chemically synthesized using D-amino acids.

## Cell lines

PGK-Luc lentiviral vector (S1 File) was produced at the Virus Core Facility of SBP. Rat glioma C6 and mouse melanoma B16F1cells from American Type Cell Culture were infected with lentivirus harboring firefly luciferase, to produce C6-Luc and B16F1-Luc lines [8, 9]. Lines were cultured in Dulbecco's-Modified Eagle + F2 medium supplemented with 10% fetal bovine serum and 100 units each/mL penicillin and streptomycin, at 37C in a humidified 5% $CO_2$ incubator. Cells were tested for mycoplasma infection using MycoAlert (Lonza Japan) and maintained under mycoplasma-free conditions.

## Binding of biotinylated D-peptides to L-MC16

To assess dTIT7 binding, wells of Sulfhydryl-BIND Surface Maleimide plates (Corning) were coated 20 hours with wild type (WT) and mutant L-MC16 in water at 4°C. After washing with PBS containing 0.02% Tween 20 (PBST), wells were blocked 1 hour with 10% superblock (Thermo) in PBST at room temperature. Biotinylated dTIT7 (10 μg dissolved in 10% superblock in PBST (1 mL) prepared as above was added to each well at 100 μl/well and incubated 30 min at room temperature for. After three PBST washes, 100 μl streptavidin-peroxidase (0.2 μg/ml) in 10% superblock in PBST containing 2% bovine serum albumin was added to each well and incubated 30 min. After three PBST washes, 100 μL of the peroxidase substrate one-step-TMB (Thermo) was added and incubated until the color developed. The reaction was stopped by adding 100 μl 2N sulfuric acid, and absorbance at 450 nm monitored using an ELISA plate reader.

## In silico conformational analysis of the ANXA1 N-terminus and dTIT7 docking

ANXA1 coordinates were obtained from the protein data bank (PDB). Both 1HM6 and 1MCX structures were derived from pig ANXA1 (89.6% sequence identity to human ANXA1 (Accession: P04083)). For the protein-protein docking structure of ANXA1, N-terminal free ANXA1 was built using MOE (Molecular Operating Environment) software ver. 2010.10 (Chemical Computing group). To obtain the dimer structure of ANXA1 with a free N-terminus, we used ZDOCK ver. 3.0.1 [18], which uses a Fast Fourier Transform-based algorithm to analyze proteins as rigid bodies during docking, searches for all possible binding orientations of a ligand along the receptor protein surface and provides docking poses ranked by Zdock scores associated with shape complementarity, desolvation and electrostatic properties. Hydrogen atoms of ANXA1 dimers calculated from Zdock were minimized using the AMBER99 force field.

## Preparation of recombinant ANXA1 protein

Recombinant, full-length ANXA1 was expressed using the baculovirus expression system [19], as described [9, 19]. Briefly, baculoviruses were prepared by recombining BacPAK6Δchi/cath baculovirus DNA with pAcP(-)-based baculovirus transfer vectors, which encode the transgene controlled by the baculovirus *p6.9* promoter. Recombinant ANXA1 protein harbored an N-terminal honeybee melittin signal peptide followed by a $His_8$-tag and the enterokinase recognition sequence DDDDR. Proteins were purified from Sf9 culture supernatants harvested 42 hours after infection using HisPur Ni-NTA resin (Pierce). Untagged ANXA1 was isolated by His-tagged enterokinase (Genscript) treatment followed by Ni-affinity chromatography. Protein concentration was determined using the BCA protein assay kit (Pierce).

## NMR measurements

NMR was analyzed in solutions consisting of 50 μM peptide(s), 10 mM $d_{11}$-Tris-HCl (pH 7.5) (Isotec Inc., IL), 150 mM NaCl, 1 mM $d_{10}$-dithiothreitol (DTT) (Isotec Inc., IL), 0.1 mM sodium 2,2-dimethyl-2-silapentane-5-sulfonate (DSS), and 5% $D_2O$. NMR spectra were recorded at 298 K on a Bruker (Germany) Avance III-500 spectrometer ($^1H$ frequency: 500.13 MHz). Chemical shifts were referenced to the peak of internal DSS. Relaxation time $T_2$ was measured using the Carr-Purcell-Meiboom-Gill sequence and analyzed with the Topspin 3.2 program (Bruker). Error levels were estimated by four repeated experiments.

## Vertebrate animal use

Mouse protocols adhered to the NIH Guide for the Care and Use of Laboratory Animals and were approved by Institutional Review Committees at National Institute of Advanced Industrial Science and Technology (AIST) and Kyoto University School of Medicine in Japan. C57BL/6 female mice (8 weeks-old) and Balb/c nu/nu female (8 weeks-old) were purchased from Charles River Laboratories Japan. C56BL/6 mice were housed under standard care, and Balb/c nu/nu mice were kept under specific pathogen-free conditions at the animal facilities of respective institutes. Animals were inspected three times a week and clinical signs were recorded. Analysis of brain tumor model mice was conducted when tumor size, as determined by photon number, was between 1 x10^4 and 1x10^7. When photon numbers exceeded 1x10^7, which approximates a volume of ~6x6x3 mm$^3$, mice were euthanized by exposing mice to saturated isoflurane gas (1~2 mL isoflurane in a 250 mL chamber) followed by cervical dislocation. No animal died before meeting criteria for euthanasia.

## Generation of brain tumor model mice

C6-Luc cells (4.8 x10$^4$ in 4 μl PBS) were injected into the brain striatum of Balb/c nu/nu mice using a stereotaxic frame as described [20]. In the same manner, B16-Luc cells (2.0 x10$^4$ cells in 2 μl PBS) were injected into the C57BL/6 brain. Five to seven days later, mice underwent imaging for luciferase-expressing tumors. To do so, 100μl luciferin (30 mg/ml PBS) was injected peritoneally, and then mice were anesthetized under isoflurane gas (20 ml/min) supplemented with oxygen (1 ml/min) and placed under a camera equipped with a Xenogen IVIS 200 imager at AIST animal facility. Photon numbers were measured for 1–10 sec or for 1 min.

## Near infra-red fluorescence whole body imaging

Each 7-mer D-peptide with an N-terminal cysteine was synthesized by GenScript (Piscataway, NJ). Peptides were conjugated with IRDye 800CW maleimide (Li-Cor) through the cysteine residue at room temperature for 2 hours according to the manufacturer's

instruction. After reverse-phase HPLC purification, the conjugate was dissolved in DMSO and 6% glucose to a final concentration of 0.2 μM. The C6-Luc brain tumor model mouse was generated in nude mice as described above. When photon number reached $5x10^4$, each IRDye-conjugated D-peptide (100 μl) was injected intravenously through the tail vein. Near infra-red fluorescence in the mouse was monitored 15 min after injection using an IVIS system and daily over 6 days.

### Conjugation of dTIT7 with a geldanamycin analogue

Procedures of Mandler *et al.* [21] were modified as follows. First, geldanamycin (GA, 100 mg) was dissolved in chloroform (18 mL), and 1, 3-diaminopropane (APA, 50 μl, molar ratio x 3.3 eq to GA) was dissolved in 2 mL chloroform. APA solution was added slowly to GA and reacted at ambient temperature under argon gas for 20 hours. Hexane (100 mL) was then added slowly to precipitate a purple product (17-APA-GA or 17-DMAG), which was filtered through a glass filter. The precipitate was solubilized in chloroform (30 mL) and conjugated immediately to *N*-maleimidobutyril oxysuccinimide ester (GMBS, 100 mg) dissolved in chloroform (10 mL) and left at ambient temperature for 60 min under argon gas. The mixture was then concentrated on a rotary evaporator and applied to silica gel for thin layer chromatography with a solvent system of chloroform: methanol (9:1, v/v). A purple band representing GMB-APA-GA was isolated and extracted from the gel with methanol. GMB-APA-GA was further purified by C18 reverse phase HPLC with an acetonitrile gradient from 40–80% in water containing 0.1% trifluoro acetic acid. HPLC-purified GMB-APA-GA was dissolved in methanol (10 mL), and C-dTIT7 peptide (equimolar to GMB-APA-GA) was also dissolved in methanol (10 mL). Both were mixed at ambient temperature for 20 hours under argon gas. The product GA-dTIT7 (1719.52 Da) was purified by HPLC. GA-dTIT7 structure was validated by MALDI TOF-MS. Control GA-C (893.44 Da), in which GA was conjugated to cysteine only, was similarly prepared.

### LC-MS/MS analysis of GA-dTIT7 in mouse serum

C57BL/6 mice (8 week-old females) (n = 6) were fasted overnight and then placed under isoflurane gas and administered a single dose of GA-dTIT7 (1 mg) dissolved with 10% taurodeoxycholate in water (200 μL) via oral gavage. Blood (50 μL) was collected from the facial vein at 0 min (pre-dose), 30 min, 60 min, 90 min and 120 min after administration using a lancet and placed into sodium heparin for plasma preparation. Then 1 μL *iso*-GA-dTIT7 (1 mg/mL in dimethylsulfoxide) was added as an internal standard to 9 μL plasma. After addition of cold acetone (40 μL), each sample was centrifuged to remove precipitates and an aliquot of supernatant was injected into an LC-MS/MS spectrometer.

### Oral administration of GA-dTIT7 to brain tumor-bearing mice

When photon numbers of B16-Luc or C6-Luc brain tumors reached $5x10^4$, oral administration of GA-dTIT7 or control GA-C was initiated. GA-dTIT7 (1719.52 Da, 2.0 mg) or GA-C (893.44 Da, 1.0 mg) was dissolved in 10 μL DMSO and diluted with 200 μL 10% taurodeoxycholate in water and then orally administered using a gavage.

### Statistical analysis

Statistical analyses were performed using GraphPad Prism software. Data sets were compared using Student's unpaired *t*-test (two-tailed). A *p* value $\leq 0.05$ was considered significant.

# Results

## Identification of linear 7-mer D-peptides by a mirror-image phage display screen

We showed previously that IF7 binds the Anxa1 N-terminal domain and that a chemically synthesized peptide representing this domain (designated MC16) was sufficient for IF7 binding [8, 9]. Here, we undertook mirror-image phage library screening to identify a protease-resistant D-type version of IF7 using synthetic D-MC16 peptide as target (Fig 1A). This procedure resulted in enrichment for several phage clones (Fig 1B–1D), many harboring a TITWPTM motif, as shown by deep sequencing (S2 File). We designated TITWPTM as TIT7; a synthetic peptide of TIT7 composed of D-amino acids was designated dTIT7.

## Binding of dTIT7 to MC16 and ANXA1 *in vitro*

Since dTIT7 interaction with the MC16 sequence within the ANXA1 N-terminal domain likely occurs when MC16 is localized to the cell membrane, we mimicked this state by coating plastic plates with MC16 peptide and then adding a solution of biotinylated dTIT7 to plates. We observed high levels of dTIT7 bound to WT MC16 in this context, with a Kd of 8.5 nM (Fig 2A). We then performed a similar analysis to assess specificity of TIT7 binding to mutant forms of MC16. That analysis indicated that dTIT7 binding affinity to MC16 mutants F7A, K9A and W11A was significantly lower than to WT MC16 (Fig 2B).

We then assessed binding of full-length ANXA1 to immobilized dTIT7 by QCM analysis, which indicated a Kd of $4.66 \times 10^{-8}$ M with ANXA1 (Fig 2C), a value comparable to that for IF7 with ANXA1 ($6.38 \times 10^{-8}$M) [9]. QCM analysis of additional D-peptides identified in our screen (namely, d-LRF7, dSPT7, dMPT7 and dLLS7) with ANXA1 showed Kd values ranging

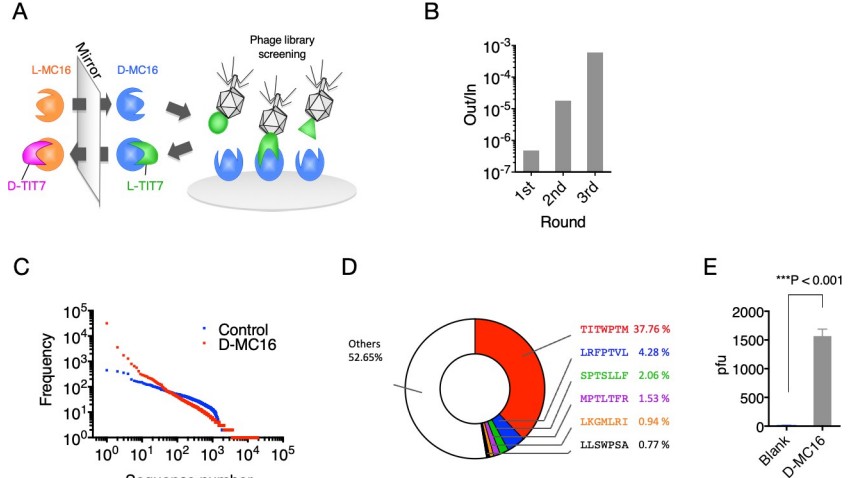

**Fig 1. Mirror-image phage library screen for MC16-binding D-peptides. A**. Strategy used to identify D-peptides using D-MC16 peptide as target. A phage library displaying L-peptides (green) is applied to the well coated with chemically synthesized D-target or D-MC16 (blue). The identified peptide sequence L-TIT7 is then chemically synthesized by D-amino acids, which should bind to natural L-target (orange). **B.** Binding efficacy of phage pools obtained after each round, as assessed by plaque-forming assays. Out/In represents the number of phage clones bound to D-MC16 (Out) per number of phage clones added to D-MC16 coated well (In). **C.** Proportion of peptides of various sequences in the third positive pool. The phage mixture was analyzed by next generation sequencing and ranked for peptide abundance (S2 File). **D.** Distribution of peptide sequences in the third positive pool. All peptide sequences are listed in S2 File. **E.** Binding of phage clones displaying the TIT7 peptide sequence to D-MC16- versus control (Blank)-coated plates.

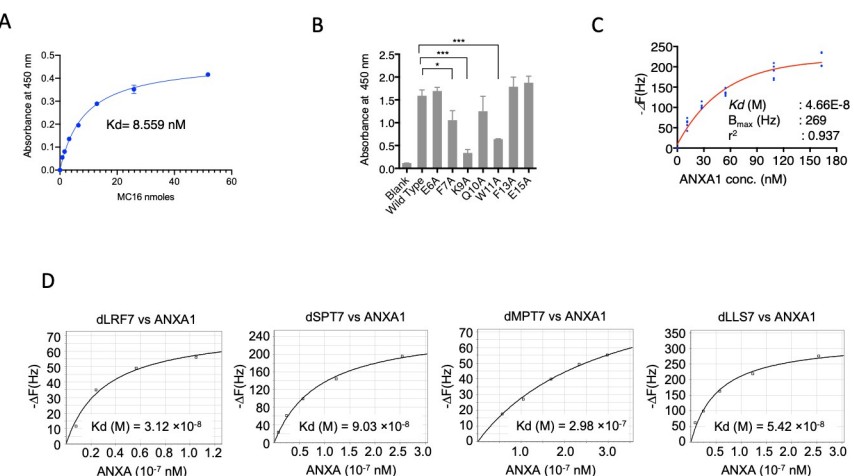

**Fig 2. Binding of dTIT7 to an ANXA1 N-terminal domain peptide or to full-length ANXA1 protein. A.** Plate binding assay of N-terminal biotinylated dTIT7 to synthetic human MC16 peptide, which represents the ANXA1 N-terminus. **B**. dTIT7 binding to human MC16 peptide and its mutants. In A and B, biotinylated dTIT7 peptide (1 μg/mL) was added to each MC16-coated plastic well and binding of peptide to MC16 was detected by a peroxidase-conjugated streptavidin and peroxidase color reaction. **C.** Binding of recombinant full-length ANXA protein to dTIT7 based on QCM analysis, which determines mass per unit area by measuring change in frequency of a dTIT7-coated sensor. **D.** Comparable QCM analysis relevant to other peptides identified in the screen.

from 3–9 x $10^{-8}$ M (Fig 2D), confirming that affinity of these peptides for ANXA1 was comparable to that of dTIT7 or IF7.

To confirm that dTIT7 and MC16 interact in solution, we analyzed a mixture of both peptides using NMR spectroscopy (Fig 3A). We observed that the spectrum of the mixture was similar but differed in key ways from the sum of respective peptides. Most prominently, a distinct peak in the mixture spectrum emerged at 0.73 ppm (Fig 3A, black arrow) but was absent in summed spectra, indicative of a significant peak shift, possibly due to a direct ordered interaction and/or to a specific conformational change. Concomitantly, we observed relative broadening of many peaks of the mixture spectrum. In Fig 3A, note that a split in the peak at 1.08 ppm (red arrow) was relatively shallower in the mixture. Peak broadening has been attributed to shortened transverse relaxation time ($T_2$) [22], which was indeed the case for the peak at 1.08 ppm (Fig 3B). Moreover, $T_2$ shortening is typically associated with an increase in molecular weight [22]. Overall, these results indicate that dTIT7 and MC16 associate in solution and do not undergo a merely disordered attraction.

We then generated a computer-simulated docking pose of dTIT7 with L-MC16 (Fig 3C). To do so, we applied the strategy used to model IF7 binding with L-MC16 [9], in which two ANXA1 N-terminal domains provide a binding pocket for dTIT7 ligand. This model estimates the free energy of binding for dTIT7 to be -5.1 kcal/mol, while that for IF7 was estimated to be -3.7 kcal/mol [9].

## dTIT7 targeting of mouse brain tumor vasculature

We then used body imaging to confirm tumor vasculature-targeting activity of dTIT7 in brain tumor model mice using a conjugate of a near infra-red fluorescent reagent IRDye 800 CW to dTIT7 peptide. IRDye-dTIT7 was injected intravenously into brain-tumor bearing nude mice, and fluorescence visualized using Xenogen IVIS imaging in real time, at various time points from 15 minutes to 144 hours (6 days) (Fig 4A). IRdye-dTIT7 targeted brain tumor and kidney and remained detectable in these locations for up to 6 days after injection. It is known that unconjugated IRDye localizes diffusely to entire mouse body.

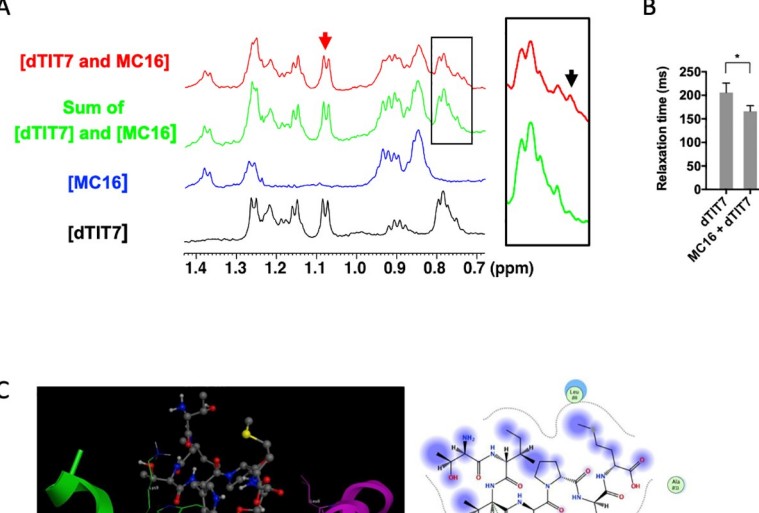

**Fig 3. NMR analysis of dTIT7 interaction with monomeric L-MC16 in solution, and computer-simulated structure model of dTIT7 bound to the ANXA1 N-terminal domain. A.** Shown are NMR spectra (methyl region) of dTIT7 (black line) or MC16 (blue line), the sum of both spectra (green line), and that of a mixture of both peptides (red line). The black arrow in the square at right indicates a peak at 0.73 ppm that emerged in the mixture spectrum, while the red arrow indicates a peak at 1.08 ppm attributable to dTIT7. **B.** Transverse relaxation time ($T_2$) of the dTIT7 peak at 1.08 ppm (red arrow in **A**) in the free state or in a mixture with MC16. **C.** Computer-simulated structural model for dTIT7 binding to the ANXA1 N-terminal domain. Previously, we proposed a model based on isothermal calorimetry suggesting that stoichiometry of IF7 binding to ANXA1 was 0.514: 1.0 [8]. Results shown here suggest that MC16 dimerization is required for dTIT7 binding. The ANXA1 dimer structure was constructed using the Zdock module for protein-protein docking [18] and the 1HM6 X-ray structure of full-length ANXA1 was added to the 1MCX core domain at residue 40 [23, 24]. The modeled structure was then hydrogenated using the Protonate 3D module in MOE. After partial charges were assigned using the AMBER99 force field [25], hydrogen atoms were minimized. The dimer structure proposed here was ranked 37th in the top 2000 structures by this program. The Alpha Site Finder module in MOE was used to identify a potential IF7 binding pocket within the dimer. The proposed model was further validated by dG scoring calculated using MOE software with GBVI/WSA, a program allowing comparison of calculated and observed energetics [26]. The dTIT7 docking pose was calculated to be -5.1 kcal/mol.

In the same model, we also tested *in vivo* tumor vasculature-targeting of additional IRDye-conjugated peptides identified in our mirror-image phage library screen, namely d-LRF7, dSPT7, dMPT7 and dLLS7, using whole body imaging. That analysis revealed signals in brain, kidney and other organs (Fig 4B). These results suggest that D-peptide sequences deduced in our screen target primarily brain tumor and kidney vasculature.

## Therapeutic activity of dTIT7-conjugated GA

Our initial criteria for peptides selection had been (1) the number of phage clones selected by phage library screen (Fig 1D), (2) binding affinity to ANXA1 (Fig 2C and 2D), and (3) *in vivo* tumor targeting activity (Fig 4). We did not further investigate dLRF7 because it did not target tumors *in vivo* (Fig 4B) and thus did not meet requirement #3. We selected dTIT7 for further investigation as this peptide satisfied all three criteria. Previously, we conjugated IF7 with GA *via* a non-cleavable linker [27], and intravenously-injected GA-IF7 suppressed tumor growth in mouse breast, prostate, lung and melanoma tumor models [8]. Here, we prepared GA-dTIT7 as we had GA-IF7 [8] (Fig 5) and determined its cytotoxicity as compared with control GA-C in C6 cells cultured *in vitro*. This assay showed the $IC_{50}$ of GA-dTIT7 and GA-C to be

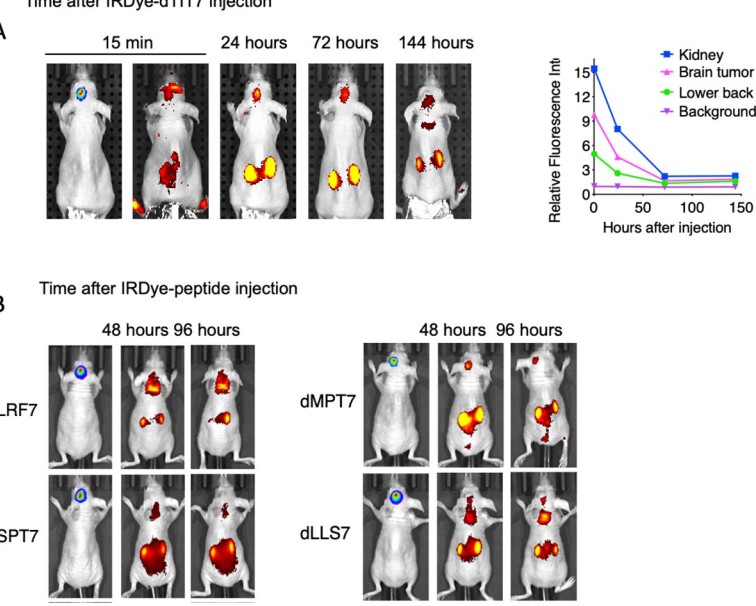

**Fig 4. Whole body image analysis of IRDye-dTIT7 in brain tumor-bearing mice. A**. Nude mice harboring C6-Luc brain tumors were injected with IRDye-dTIT7 through the tail vein. Whole body imaging of infra-red fluorescence was monitored using an IVIS imager. Right graph shows quantitative analysis of infra-red fluorescence signals in mice shown at left. **B.** Comparable whole body imaging for IRDye-conjugated dLRF7, dSPT7, dMPT7 and dLLS7 peptides.

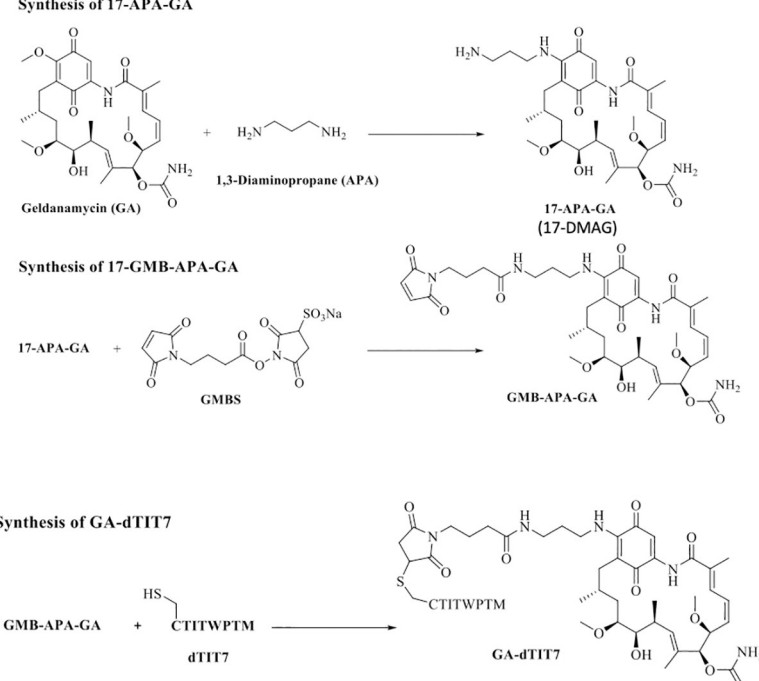

**Fig 5. Steps in synthesis of GA-dTIT7.** Procedures described by Mandler *et al.* [21] were modified as described in Materials and Methods. Note that 17-APA-GA is also known as 17-DMAG [28].

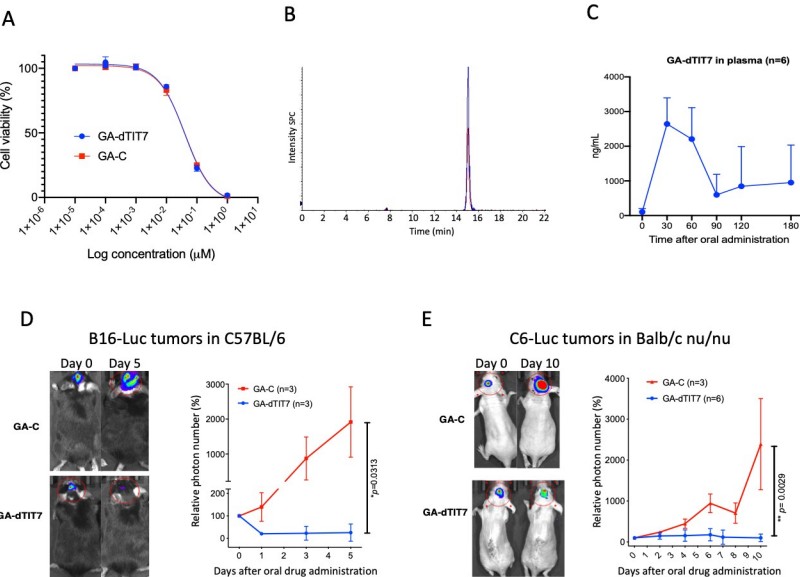

**Fig 6. Therapeutic effect of orally-administered GA-dTIT7 on brain tumors. A.** Mouse melanoma B16 cells were treated with reagents shown at indicated concentrations, cultured 2 days, and assessed for viability using a CellTiter Glo (Promega) assay. The IC$_{50}$ of each reagent was determined using GraphPad Prism software. **B.** Quantitative analysis of GA-dTIT7 in mouse plasma by LC-MS/MS. Plasma from GA-dTIT7-injected C57BL/6 female mice (9 μL) were combined with 1 μL GA-*iso*dTIT7 (1.0 μg), immediately mixed with 40 μL cold acetone, and then centrifuged to remove precipitates. The supernatant was then applied to LC-MS/MS, and eluates monitored by m/z 1725 for GA-*iso*dTIT7 (blue) and m/z 1719 for GA-dTIT7 (red). **C.** GA-dTIT7 levels in plasma from mice-orally administered GA-dTIT7. Each C57BL/6 mouse was administered 1 mg GA-dTIT7. GA-dTIT7 levels were determined by LC-MS/MS, as shown in B. **D.** B16-Luc cells were injected into the brain of C57BL/6 mouse and tumor growth was monitored by IVIS imaging. When photon number reached 2 x10^4 (approximately 5 days after inoculation), GA-dTIT7 (1.16 μmoles or 2 mg) or the molar equivalent GA-C (control) diluted with 10% TDC (200 μl) was orally-administered daily for 5 days. Left panels show representative control and experimental mice imaged 0 and 5 days after drug administration. Photon number is quantified at right. **E.** C6-Luc cells were injected into the brain of C57BL/6 mice and tumor growth monitored by IVIS imaging. When photon number reached 2 x10^4 (approximately 10 days after C6-Luc cell inoculation), GA-dTIT7 and control GA-C were orally administered daily for 10 days. Left panels show representative control and experimental mice imaged 0 and 10 days after drug administration. Photon number is quantified at right. In these graphs, error bars denote means ± SEM. Statistical analysis was assessed by Student's t-test.

0.396 nM and 0.410 nM, respectively (Fig 6A), suggesting that GA-C serves for the control of GA-dTIT7.

Previously, we found that intravenously-injected GA-IF7 at 6.5 μmoles/kg suppressed growth of melanoma, lung carcinoma, prostate cancer, and breast cancer models in mouse [8]. Since it is known that the GA analogue 17-DMAG, which is highly similar to the GA moiety of GA-dTIT7 (Fig 5), is orally-administrable [28], we asked whether GA-dTIT7 administered orally would enter the circulation by assessing gut-to-blood GA-dTIT7 transport using quantitative LC-MS/MS analysis of isotopically-labeled dTIT7 (*iso*dTIT7). The molecular weight of GA-*iso*dTIT7, in which the last methionine residue contains $^{13}$C and $^{15}$N, is 1725.60 Da, while that of the internal standard, GA-dTIT7, is 1719.6 Da (Fig 6B). For this analysis, we dissolved GA-dTIT7 in 10% taurodeoxy-cholate (TDC) in water to enhance drug transport from the digestive tract to the circulation. Following oral administration of 1 mg (0.58 μmoles) GA-dTIT7 per mouse, we combined plasma samples from mice with GA-*iso*dTIT7. After removal of proteins by precipitation with cold acetone, we subjected the supernatant to LC-MS/MS analysis to determine the quantity of GA-dTIT7 in plasma, using GA-*iso*dTIT7 as an internal control. This analysis showed a time-dependent increase in GA-dTIT7 in mouse plasma, peaking at 30 min after oral administration (Fig 6C). At that peak, drug concentration was 2.62 ± 0.69 ng /mL, or 1.52 nM.

Next, we tested the therapeutic effect of orally-administered GA-dTIT7 *in vivo*. We had previously shown that IF7-SN38 overcame the BBB and suppressed brain tumor growth in model mice [9]. To determine whether GA-dTIT7 functioned similarly, we established B16-Luc tumors in brains of C57BL/6 mice and monitored tumor growth by photon number produced by luciferase using IVIS imaging. When photon number reached $1 \times 10^4$, we orally administered GA-dTIT7 (1 mg or 0.58 μmoles) in 200 μL in 10% TDC in water daily for 7 days but did not observe suppression of tumor growth. However, when we doubled the GA-dTIT7 dose to 2 mg and orally-administered the drug daily for 5 days, imaging revealed significant suppression of tumor growth in GA-dTIT7-treated mice, while tumors continued to grow in control mice that had received 1 mg GA-C (the molar equivalent of GA-dTIT7) daily for 5 days (Fig 6D). Comparable analysis using C6-Luc brain tumor models in nude mice revealed tumor growth suppression by oral administration of GA-dTIT7 but not control GA-C (Fig 6E). These results provide proof-of-concept that orally-administered GA-dTIT7 suppresses brain tumor growth *in vivo* in mice.

## Discussion

Here we used a mirror-image peptide display strategy [15] to identify a series of linear 7-mer D-peptides using the ANXA1 $NH_2$-terminal domain peptide as target (Fig 1A). Because this strategy uses a chemically synthesized receptor made of D-amino acids, the application is limited to proteins in which a synthetic version of the peptide functions as receptor for the protein of interest. Nonetheless, this strategy has been successfully applied to develop therapeutic D-peptide modulators of the tyrosine kinase SH3 domain [16] and inhibitors of amyloid beta aggregation in Alzheimer's disease [17]. In both cases, each D-target conformed to a unique stereo-specific structure and provided a binding pocket for L-peptides displayed on the phage. In our study, we also exploited the fact that a chemically-synthesized peptide (MC16) representing the ANXA1 $NH_2$-terminal domain served as receptor for IF7 [9]. Although MC16 is considered too short and flexible in solution to form a stable 3-D structure, IF7/MC16 interactions were detected in our binding assays, which included a plate binding assay, fluorescence correlation spectroscopy, and QCM [9]. Indeed, D-peptides targeting D-MC16 also bound to L-MC16 and full-length ANXA1 protein (Figs 2 and 3).

Others have reported a D-peptide IF7 alternative designated retro-inverso IF7 (RIF7), in which the reverse IF7 sequence was synthesized using D-amino acids [12]. They reported that when RIF7 was conjugated to red fluorescent 5-carboxytetramethylrhodamine (TMR) and injected intravenously into a pulmonary cancer model mouse, TMR-RIF7 targeted the lung tumor and exhibited prolonged stability compared to TMR-IF7 [12]. That study showed that TMR-IF7 and TMR-RIF7 also targeted normal organs. Such non-specific organ targeting is likely due partially to TMR. Thus far no one has reported a therapeutic effect of a RIF7-conjugated drug.

In this study, we showed that IRDye-dTIT7 can be used for brain tumor detection by whole body imaging (Fig 6). As unconjugated IRDye does not target normal mouse organs, the kidney targeting of IRDye-peptides may be caused by a unique structure created by IRDye/dTIT7 conjugates that binds to an unknown receptor expressed on the vasculature surface of normal kidney. This hypothesis is supported by our previous study showing that intravenously injected fluorescent Alexa488-IF7 targeted brain tumors but not kidney [9].

Currently, tumors are often diagnosed by positron emission tomography (PET) scans utilizing radioactive [18]F glucose or FDG. Despite the highly specific tumor vasculature targeting activity of IF7, our attempts to conduct PET with IF7 were not successful (data not shown), although others have shown detectable, though limited, tumor imaging with IF7 [29–31]. We

emphasize, however, that whole body imaging of IRDye-conjugated dTIT7 indicated clear brain tumor targeting (Fig 4). Compared with other organ systems, FDG-PET imaging of brain presents unique challenges due to high background glucose metabolism in normal gray matter [32]. We consider that D-peptides identified here warrant further testing in imaging of brain tumors.

Although we had anticipated that intravenously-injected GA-dTIT7 would exhibit anti-tumor activity *in vivo*, we did not observe therapeutic activity of dTIT7-conjugated drugs following intravenous injection, suggesting that either higher dosages of GA-dTIT7 or different drug formulation may be required. Relevant to the latter, we have found that detergents significantly alter IF7-SN38 therapeutic efficacy: formulation with 10% Solutol in water significantly reduced the effective dosage against brain tumors [9]. Future pharmacokinetic studies should address these issues in the case of GA-dTIT7 following intravenous injection.

GA analogues 17-AAG and 17-DMAG are reportedly potent anti-cancer agents with less toxicity than the parental drug GA. However, several clinical trials with GA analogues indicated toxicity too high to proceed beyond a phase II trial [33, 34]. Nonetheless, pre-clinical and clinical studies of the GA analogue 17-DMAG showed that it is orally-administrable [28, 35]. We found that GA-dTIT7 (Fig 5) is also orally-administrable and suppressed tumor growth in mouse brain tumor models (Fig 6D and 6E). We predict that GA-dTIT7 would be stable *in vivo*, due to the esterase-resistant linker and protease resistance of dTIT7. Efficacy of GA-dTIT7 gut-to-blood transport was low here (Fig 6C), and future studies should address this concern to strengthen the clinical relevance of this drug.

Cancer treatments are increasingly expensive due to development of sophisticated diagnostics and therapies. Our drug, which consists of a short peptide plus an anti-cancer reagent, can be chemically synthesized cost-effectively. Given that ANXA1 is an extremely specific tumor vasculature surface marker [2], and IF7-conjugated anti-cancer drugs have significant effects on subcutaneous and brain tumors [8, 9, 36], a drug conjugated to an ANXA1-binding peptide should eradicate tumors effectively at low dosage with few side effects. Finally, orally-administrable drugs would be advantageous in economically disadvantaged societies that lack infrastructure required for costly treatment. As clinical trials with tumor vasculature-homing peptides are beginning, we will soon be able to evaluate efficacy of these strategies in cancer patients. Further development of peptide-conjugated drugs could reveal strong candidates for clinical applications to treat intractable cancers.

## Supporting information

**S1 File. Nucleotide sequence of the PGK-Luc vector.**
(PDF)

**S2 File. Nucleotide and peptide sequences obtained by NGS (next generation sequencing) of the third positive phage pool.**
(XLS)

**S3 File. Whole body imaging of IRDye-dTIT7 on brain tumor bearing mice.**
(PDF)

## Acknowledgments

The funders had no role in study design, data collection and analysis, decision to publish, or preparation of the manuscript. We thank Dr. Elise Lamar for editing the manuscript and Mrs. Hisae Okuhara for clerical/administrative assistance.

## Author Contributions

**Conceptualization:** Michiko N. Fukuda.

**Data curation:** Motohiro Nonaka, Hideaki Mabashi-Asazuma, Tomoya O. Akama, Masato Nagaoka, Toshio Sasai, Itsuko Kimura-Takagi, Yoichi Suwa, Chizuko Nishizawa-Harada, Michiko N. Fukuda.

**Formal analysis:** Masato Nagaoka, Michiko N. Fukuda.

**Funding acquisition:** Donald L. Jarvis, Michiko N. Fukuda.

**Investigation:** Motohiro Nonaka, Donald L. Jarvis, Kazuhiko Yamasaki, Tomoya O. Akama, Masato Nagaoka, Toshio Sasai, Itsuko Kimura-Takagi, Takashi Yaegashi, Chun-Teng Huang, Chizuko Nishizawa-Harada, Michiko N. Fukuda.

**Methodology:** Motohiro Nonaka, Hideaki Mabashi-Asazuma, Donald L. Jarvis, Kazuhiko Yamasaki, Tomoya O. Akama, Masato Nagaoka, Toshio Sasai, Itsuko Kimura-Takagi, Yoichi Suwa, Takashi Yaegashi, Chun-Teng Huang, Chizuko Nishizawa-Harada, Michiko N. Fukuda.

**Resources:** Hideaki Mabashi-Asazuma, Masato Nagaoka, Yoichi Suwa, Takashi Yaegashi, Chun-Teng Huang.

**Software:** Motohiro Nonaka, Donald L. Jarvis, Kazuhiko Yamasaki, Masato Nagaoka, Toshio Sasai, Yoichi Suwa.

**Supervision:** Masato Nagaoka, Michiko N. Fukuda.

**Validation:** Motohiro Nonaka, Hideaki Mabashi-Asazuma, Donald L. Jarvis, Kazuhiko Yamasaki, Tomoya O. Akama, Masato Nagaoka, Toshio Sasai, Yoichi Suwa, Takashi Yaegashi, Chizuko Nishizawa-Harada, Michiko N. Fukuda.

**Visualization:** Motohiro Nonaka.

**Writing – original draft:** Motohiro Nonaka, Michiko N. Fukuda.

**Writing – review & editing:** Michiko N. Fukuda.

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
