## [Decision Letter · Decision Letter 0]

10 Nov 2020

PONE-D-20-31321

Development of an orally-administrable tumor vasculature-targeting therapeutic using annexin A1-binding D-peptides

PLOS ONE

Dear Dr. Fukuda,

Thank you for submitting your manuscript to PLOS ONE. After careful consideration, we feel that it has merit but does not fully meet PLOS ONE’s publication criteria as it currently stands. Therefore, we invite you to submit a revised version of the manuscript that addresses the points raised during the review process.

We look forward to receiving your revised manuscript.

Kind regards,

Yu-Hsuan Tsai

Academic Editor

PLOS ONE

2. At this time, we request that you  please report additional details in your Methods section regarding animal care, as per our editorial guidelines:

(i) Please state the source and number of mice for all strains used in the study.

(ii) Please describe the care received by the animals, including the frequency of monitoring and the criteria used to assess animal health and well-being.

Thank you for your attention to these requests.

3. Thank you for providing the following information regarding your tumor model "Experiments of brain tumor model mouse were conducted when tumor size determined as photon number was between 1 x10^4 and 1x10^7. When brain tumor grew more than 1x10^7, the mouse was euthanized by placing the animal under saturated isoflurane gas (1~2 mL isoflurane in 250 147 mL chamber) followed by cervical dislocation. No animal died before meeting the criteria for euthanasia."

In the manuscript methods, please provide the tumor volume in mm3 that equates to 1x10^7 photons.

4. In your Methods section, please provide additional details regarding the cell lines used in your study and ensure you have described the source. Please also provide additional information about each of the cell lines used in this work, including any quality control testing procedures (authentication, characterisation, and mycoplasma testing).

For more information regarding PLOS' policy on materials sharing and reporting, see https://journals.plos.org/plosone/s/materials-and-software-sharing#loc-sharing-materials, and for more information on PLOS ONE's guidelines for research using cell lines, see https://journals.plos.org/plosone/s/submission-guidelines#loc-cell-lines

5. Please note that PLOS does not permit references to “data not shown.” Authors should provide the relevant data within the manuscript, the Supporting Information files, or in a public repository. If the data are not a core part of the research study being presented, we ask that authors remove any references to these data.

6. We note that this submission includes NMR spectroscopy data. We would recommend that you include the following information in your methods section or as Supporting Information files:

a) For each individual experiment, please list the reference standard and the temperature.

b) A list of the chemical shifts for all compounds characterised by NMR spectroscopy, specifying, where relevant: the chemical shift (δ), the multiplicity and the coupling constants (in Hz), for the appropriate nuclei used for assignment.

c)The full integrated NMR spectrum, clearly labelled with the compound name and chemical structure.

We also strongly encourage authors to provide primary NMR data files, in particular for new compounds which have not been characterised in the existing literature. Authors should provide the acquisition data, FID files and processing parameters for each experiment, clearly labelled with the compound name and identifier, as well as a structure file for each provided dataset. See our list of recommended repositories here: https://journals.plos.org/plosone/s/recommended-repositories

7. Please provide the source of the luciferin used in your study.

8. We note that you describe B16-Luc brain tumors in line 189, but only describe the injection of C6-Luc cells into mice on line 146. Please ensure all experimental procedures in mice are reported in your Methods.

9. Please ensure your Methods and reagents are described in sufficient detail for another researcher to reproduce the experiments described. Specifically, please include

a) the peptide sequences of the peptides purchased and

b) the nucleotide sequence of the PGK-Luc lentiviral vector for your study as supplementary data.

10. Thank you for stating the following in the Acknowledgments Section of your manuscript:

'This study was supported by an institutional grant LEAD at National Institute of Advanced Industrial Science and Technology, by the Project for Cancer Research and Therapeutic Evolution (PCREATE) from Japan Agency for Medical Research and Development (AMED) to MNF, by P41 GM103390 and P41 RR005351. NM is a recipient of a Research Grant for Young Japanese Scientists from The Nakajima Foundation.

'The funders had no role in study design, data collection and analysis, decision to publish, or preparation of the manuscript.'

Reviewers' comments:

Reviewer's Responses to Questions

**Comments to the Author**

1. Is the manuscript technically sound, and do the data support the conclusions?

Reviewer #1: Yes

Reviewer #2: Yes

Reviewer #3: Yes

2. Has the statistical analysis been performed appropriately and rigorously? 

Reviewer #1: No

Reviewer #2: Yes

Reviewer #3: Yes

3. Have the authors made all data underlying the findings in their manuscript fully available?

Reviewer #1: Yes

Reviewer #2: Yes

Reviewer #3: Yes

4. Is the manuscript presented in an intelligible fashion and written in standard English?

Reviewer #1: No

Reviewer #2: Yes

Reviewer #3: Yes

5. Review Comments to the Author

Reviewer #1: Below is my criticism of the manuscript;

52-53 There should be elaboration here into the nature of origin of the 7mer IF7 is at lays the foundation of this work. Was this discovered by screening? Is it derived from natural binding protein?

58-59 This statement should be explained by the relation of transcytosis to the blood brain barrier – otherwise this lacks context for the non-expert.

70 Literature indicates that residues 2-26 of ANXA1 of the N-terminal domain serves as the second membrane binding site, with accompanying solution NMR and CD to show alpha helicity. It is unclear why residues 1-15 were chosen specifically.

223-227 TIT7 sequence was the focal point for most of the experiments due to the majority enrichment in the phage clones Fig1D. dTIT7 binds to ANXA1 with a Kd of 46.4 nM. Fig 2D shows dLRF7 has tighter binding at 31.2 nM. It is unclear why dTIT7 was taken forward at this point and dLRF7 was disregarded. Plus, error analysis is needed.

270 Since the activity of the D-peptide drug conjugates is significantly less than the original L-IF7 conjugates – the utility of the D-peptide vehicles presented in this work could be demonstrated with pharmacokinetic analysis.

Dissociation constants of Fig 2A and C could be larger as it is difficult to read.

229 NMR at 500 MHz frequency does not provide sufficient resolution to make a strong argument of solution-phase binding. The differences between the spectra are not clear. Also, there is no evidence that MC16 adopts an alpha helix in solution. Ideally, the work would include CD spectra of MC16 and D-MC16 under similar conditions to literature NMR (50% TFE/Water or 10mM SDS) and similar conditions used for any analysis of solution-state binding. Otherwise, any indicated binding between dTIT7 and the MC16 can be argued as non-specific interaction.

246-257 D-TIT7 accumulates in the kidney vasculature at a significantly higher level than the brain tumour. Is this as indictment of using Annexin A1 as a surface marker in malignant tumour vasculatures. Or is this a question of off target effects of the peptide-conjugates. An explanation is not offered in the discussion.

Overall, the work is original and demonstrates a clear application of utilising short protein domains to identify D-peptide binders for drug delivery leads – albeit with some concerns as to whether MC16 is a good enough model of the N-terminal binding domain in this case. The points made on the introduction should be addressed prior to publication, as it impacts the communication of the presented work.

Lastly, in the introduction, the authors need to explain why D-peptides, rather than cyclic L-peptides which have also been suggested to be biologically stable, are used.

I think this manuscript is suitable for publication in PLoS one.

Reviewer #2: The paper by Nonaka et al., on “Development of an orally administrable tumor vasculature targeting therapeutic using annexin A1 binding D-peptides” deals with an important step in tumor theranostic and this is through orally administrable drug/peptide. It is well thought study with profound inside however there are some missing links. Hence, I recommend it to publish with minor revision considering the impact and importance of the outcome.

Minor comments:

(1) What is the genesis of selecting ANXA1-binding D-type peptide apart from getting this from Phage library? Is there availability of D-type peptides in tumor related cases; it should be clearly brought in in “introduction”?

(2) Why linear 7-mer peptides library were chosen? Can one use shorter or longer length? What if peptides are cyclic?

(3) In Figures 3 as well as in results section, authors mention about IF7 binding to Anxa1 dimer. What is stoichiometry of binding and does binding break the dimerization or leads to multi-merization?

(4) From Figure 1D, there are ~53% other peptides? Are they just random or have homology to the prominent ones; should not be they considered for study?

Reviewer #3: In this manuscript, Nonaka et al. developed an orally-administrable D-peptide-drug conjugate targeting ANXA1 receptor expressed on cancer cells. They first identified peptide sequences that bind to a D-peptide derived from N-terminal domain of ANXA1 through mirror-image phage display. Interaction between D-peptides and L-ANXA1 peptide and protein was then confirmed by ELISA, QCM and NMR. The authors tested the localization of selected D-peptides in mouse whole body using fluorescence imaging and confirmed targeted localization. Although intravenous injection of D-peptide-drug conjugate was not effective probably due to an insufficient dose, they finally observed therapeutic effect of GA-dTIT7 through oral administration.

The results shown in this manuscript is interesting, convincing, and suitable for the readers of PLOS one. Therefore, this report would be acceptable after the authors addressed specific comments and questions shown below.

1) In Figure 4, there are no control experiments using some D-peptide with random sequence. To clarify the effectiveness of selected D-peptide for targeted localization, the authors should add a result of the control experiment, if possible.

2) In the caption of Figure 6, “B.” should be bold. “C. B16-Luc cells,,,” and “D. C6-Luc cells,,,” should be “D. B16-Luc cells,,,” and “E. C6-Luc cells,,,”, respectively.

3) How much is the diversity of phage library used in this study? The authors should clarify it in the text.

6. PLOS authors have the option to publish the peer review history of their article (what does this mean?). If published, this will include your full peer review and any attached files.

Reviewer #1: No

Reviewer #2: No

Reviewer #3: No

---

## [Author Response · Author response to Decision Letter 0]

3 Dec 2020

Responses are described in the cover letter.

---

## [Editor Report · Decision Letter 1]

7 Dec 2020

PONE-D-20-31321R1

Development of an orally-administrable tumor vasculature-targeting therapeutic using annexin A1-binding D-peptides

PLOS ONE

Dear Dr. Fukuda,

Thank you for submitting your manuscript to PLOS ONE. After careful consideration, we feel that it has merit but does not fully meet PLOS ONE’s publication criteria as it currently stands. Therefore, we invite you to submit a revised version of the manuscript that addresses the points raised during the review process.

We look forward to receiving your revised manuscript.

Kind regards,

Yu-Hsuan Tsai

Academic Editor

PLOS ONE

Additional Editor Comments:

1. Is there a difference between Anxa1 and ANXA1? If not, please unify the format.

2. Please add explanation about why dLRF7 was not further investigated into the discussion. It is worthwhile to elaborate the peptide selection criteria as well.

3. Figure resolution is quite bad in the pdf file. Please ensure that they are of sufficient resolution in the final version.

4. Explanation of why IRDye-conjugated dTIT7 targeted the kidney should be in Discussion. It is also worthwhile to propose experiments on verify possible mechanism.

5. It is still not entirely clear why D-peptides were chosen over L-peptides for this study. Could you please further elaborate your reason?

6. Line 79, please provide more background info about retro-inverso. This is not readily understandable by researchers slightly outside the field.

7. Regarding negative control of Fig 4, what is the localization of IRDye? Also, are there biological replicates? If so, please include them in the SI.

8. Could you please provide a calculation on how the diversity of the phage library is calculated? Was each amino acid coded by NNN or NNK?

9. Lastly, the manuscript will benefit from language editing or thorough check of grammar and the use of language.

---

## [Author Response · Author response to Decision Letter 1]

15 Dec 2020

Responses are submitted as a separate document as response to the editor.

---

## [Editor Report · Decision Letter 2]

17 Dec 2020

Development of an orally-administrable tumor vasculature-targeting therapeutic using annexin A1-binding D-peptides

PONE-D-20-31321R2

Dear Dr. Fukuda,

We’re pleased to inform you that your manuscript has been judged scientifically suitable for publication and will be formally accepted for publication once it meets all outstanding technical requirements.

Kind regards,

Yu-Hsuan Tsai

Academic Editor

PLOS ONE

---

## [Editor Report · Acceptance letter]

21 Dec 2020

PONE-D-20-31321R2 

Development of an orally-administrable tumor vasculature-targeting therapeutic using annexin A1-binding D-peptides 

Dear Dr. Fukuda:

I'm pleased to inform you that your manuscript has been deemed suitable for publication in PLOS ONE. Congratulations! Your manuscript is now with our production department. 

Kind regards, 

on behalf of

Dr. Yu-Hsuan Tsai 

Academic Editor

PLOS ONE